# Clinical and Eco-Epidemiological Aspects of a Novel Hyperendemic Area of Paracoccidioidomycosis in the Tocantins-Araguaia Basin (Northern Brazil), Caused by *Paracoccidioides* sp.

**DOI:** 10.3390/jof8050502

**Published:** 2022-05-12

**Authors:** Alessandra G. Krakhecke-Teixeira, Danielle H. Yamauchi, Alexsandra Rossi, Herdson R. de Sousa, Hans G. Garces, Joaquim L. Júnior, Antônio O. S. Júnior, Maria Sueli S. Felipe, Eduardo Bagagli, Heitor F. de Andrade, Marcus de M. Teixeira

**Affiliations:** 1Tocantins Department of Public Health, Palmas 77015-007, TO, Brazil; agkrakhecke@gmail.com (A.G.K.-T.); alexsarossi@gmail.com (A.R.); 2Institute of Tropical Medicine, University of São Paulo, São Paulo 05403-000, SP, Brazil; hfandrad@usp.br; 3Department of Chemical and Biological Sciences, São Paulo State University “Júlio de Mesquita Filho”, Botucatu 18618-687, SP, Brazil; danihyamauchi@gmail.com (D.H.Y.); atiestorage@gmail.com (H.G.G.); ebagagli@gmail.com (E.B.); 4Núcleo de Medicina Tropical, Faculty of Medicine, University of Brasília, Campus Universitário Darcy Ribeiro, Brasília 70910-900, DF, Brazil; herdson10@gmail.com; 5Oswaldo Cruz Foundation (Fiocruz), Brasília 70904-130, DF, Brazil; joaquim.unir@hotmail.com; 6Medical School, Federal University of Tocantins, Araguaína 77402-970, TO, Brazil; antonio.ojunior@ebserh.gov.br; 7Graduate Program in Genomic Sciences and Biotechnology, Catholic University of Brasília, Brasília 70790-160, DF, Brazil; msueliunb@gmail.com

**Keywords:** Tocantins-Araguaia basin, *Paracoccidioides brasiliensis*, *Paracoccidioides lutzii*, paracoccidioidomycosis, Northern Brazil

## Abstract

Paracoccidioidomycosis (PCM) is the most prevalent systemic mycosis in Brazil. The disease is caused by dimorphic fungi nested within the Paracoccidioides genus. We described 106 PCM cases (47.1 cases/year) at the Tropical Diseases Public Hospital of Tocantins State. PCM was prevalent in males and rural workers over 50 years; the chronic pulmonary form predominated in 67% of cases. The male-to-female ratio was 2.65:1, with more women affected than other endemic regions of Brazil. Urban or indoor activities were reported in women and are ascribed to disease urbanization. qPCR-based assays confirmed the identification of Paracoccidioides DNA in 37 biological specimens. Paracoccidioides sp. DNA was found in 53% of the environmental samples, suggesting autochthonous infections. Therefore, the Tocantins-Araguaia basin must be considered a novel hyperendemic area of PCM in Brazil, reinforcing the importance of including PCM as a notifiable disease, requiring specific diagnosis and health measures.

## 1. Introduction

Mycotic infections are on the rise worldwide, and the emergence and re-emergence of fungal pathogens are related to climate change, deforestation, agricultural practices, biodiversity loss, human occupation, and uncontrolled use of immunosuppressive drugs [1]. Paracoccidioidomycosis (PCM) is the most prevalent systemic mycosis in Latin America, with most cases occurring in Brazil, Colombia, and Venezuela, affecting specific social segments such as rural workers or farmers [2]. *Paracoccidioides brasiliensis*, *P. americana*, *P. restrepiensis*, *P. venezuelensis*, and *P. lutzii*, are the etiological agents of PCM [3,4]. However, *P. loboi* and *P. cetii* were recently added as the causative agents of paracoccidioidomycosis loboi (human pathogen) and paracoccidioidomycosis ceti (dolphin pathogen), respectively [5]. *Paracoccidioides* are thermodimorphic fungi found in nature as mycelia, producing infectious conidia. Upon the inhalation by susceptible hosts, the infection is established, and the fungus shifts its morphology to pathogenic multibudding yeasts [6]. The natural infection caused by *Paracoccidioides* is asymptomatic or presents subclinical symptoms in approximately 95% of cases [7]. However, affected individuals develop a deep systemic mycosis responsible for a high morbidity and mortality rate in Latin America [8,9]. Two main forms of the disease are observed: The acute/subacute form affects children and adolescents primarily, presenting intra-abdominal lymphadenomegaly, hepatosplenomegaly, and skin oral and intestinal mucosa lesions. The chronic form is a long-term disease restricted to the lungs and upper respiratory tract and may disseminate to the buccal mucosa, sinus, and other parts of the body; this form of the disease accounts for 80% to 95% of cases and is frequently observed in patients after 30 years [2].

PCM was responsible for 3181 deaths in Brazil between 1980 and 1995, representing an annual average of 198.81 deaths and a mortality rate of 1.45 per one million inhabitants. The mortality rate is the highest among the systemic mycoses and is the eighth highest cause of mortality among chronic parasitic diseases, exceeding schistosomiasis and leishmaniasis [9]. These statistics demonstrate that PCM has a high prevalence, affects mainly low-income populations, lacks awareness and public health planning, and is considered a neglected disease [10,11]. Furthermore, the absence of compulsory notification coupled with a lack of precise and fast diagnostic tools impedes the accurate determination of the prevalence and incidence of PCM [12]. In recent decades, remarkable changes have been observed in the epidemiology of PCM, the demographics of the affected population, and the geographical distribution of the disease following the expansion of the agricultural frontiers and an extensive urbanization process [2,13]. Furthermore, environmental factors arising from opening new agricultural lands with deforestation, especially in the midwestern and northern areas of Brazil (including the Brazilian savanna and the Amazon biomes), have also contributed to the current PCM landscape [10,14].

PCM was described as the most common systemic fungal infection in Araguaína, Tocantins state, between 1998 and 2001 [15]. This region is strategically important to the epidemiology of PCM once it is located in the Tocantins-Araguaia basin, one of the most affected areas in the past three decades due to extensive cattle farming and agriculture. Moreover, Araguaína and surrounding municipalities are located in the confluent ecozone MATOPIBA (Maranhão, Tocantins, Piauí, and Bahia states) and are considered the last Brazilian agricultural frontier, presenting patches of savanna and Amazon, which are crucial to precisely determine species of *Paracoccidioides* and disease phenotypes occupying both biomes.

More recently, a new species of *Paracoccidioides* was described in Midwestern Brazil, named *P. lutzii*, with distinct morphology and antigen composition but with similar clinical manifestations [4,16,17,18]. Therefore, understanding the genotypic background of *Paracoccidioides* strains from patients from the Tocantins-Araguaia basin is needed to avoid missing false-negative serological tests. The present study aimed to investigate patients diagnosed with PCM and treated at the Public Hospital for Tropical Diseases of Tocantins. We have found 106 new cases of paracoccidioidomycosis disease in 27 months (47.1 cases/year), ranking the Northern Tocantins state and surrounding areas as hyperendemic for paracoccidioidomycosis. Furthermore, we have identified the presence of *Paracoccidioides* sp. DNA in 37 biological specimens using a qPCR-based test targeting the *gp43* and *hsp70* loci. We have also used a nested PCR assay to determine the presence of the fungus in the soil in this endemic area, suggesting that local infections are likely to occur.

## 2. Materials and Methods

### 2.1. Clinical Cases and Diagnostics

We analyzed clinical data and samples from 240 patients with clinical suspicion of PCM treated at the Public Hospital for Tropical Diseases of Tocantins (PHTD), Hospital regional, and Hospital Don Orioni in Araguaína from October 2010 to December 2012, after approval by the Ethics in Research Tropical Medicine Foundation of Tocantins (number 164). A total of 327 biological samples such as sputum, Broncho-Alveolar Lavage (BAL), and tissues biopsies were analyzed by direct examination with 30% KOH on an optical microscope with 10× and 40× objectives [19]. These samples were seeded onto Sabouraud dextrose and Mycosel agar and incubated at 37 °C for up to 60 days in an incubator [6]. Yeast cells were stained with acridine orange and visualized on an LSM 510 META inverted confocal microscope (Zeiss). In addition, clinical and occupational data were recovered from clinical records. Data were analyzed using Epi-Info, and a chi-squared test was applied to test significance between two variables (CDC, version 3.5.1, Atlantic City, NJ, USA).

### 2.2. Molecular Diagnostics

DNA was extracted from 51 clinical samples and one *Paracoccidioides* sp. isolate using the QIAamp DNA Mini Kit^®^ (Qiagen, Germany) according to manufacturer’s instructions, aiming to identify *Paracocidioides* DNA using a qPCR-based test. We used the *gp43* assay: (5′-TTCCCAAAACGGCTTCGA-3′), (5′-TGTCACCCTTTTGCCAGTTG-3′), probe 5′-VIC_ACAGCGGTCACCGTGGCGC_TAMRA-3′, which generates a 61 bp sequence [20], and the *hsp70* assay (5′-CGTATCCCCCGCATCCA-3′), (5′-TGGCTCCTTGCCGTTGA-3′), probe (3′-FAM_AAGCTTGTATCCGACTTC_MGB-3′), which produces a 54 bp amplicon. The *hsp70* qPCR assay was developed based on the *hsp70* sequence (GenBank: AF386787.1) using the Primer Express Software version 3.0 tools (Applied Biosystems^®^). Real-time PCR reactions were conducted on an Applied Biosystems^®^ 7500 instrument in a final volume of 25 µL containing 3 mM MgCl_2_, 0.5 µM of each primer, 0.6 µM probe, and 5 µL of DNA template using the SYBR Green Master Mix (ThermoFisher, Waltham, MA, USA). Each experiment included as positive controls the *P. lutzii* Pb01 and 8334 strains, the *P. brasiliensis* Pb339 strain, and as negative controls *Candida* sp. and *Fusarium* sp. The conditions of the real-time PCR were: 95 °C for 10 min and 45 amplification cycles (15 s at 95 °C, 30 s at 56 °C, and 5 s at 72 °C). Samples that presented amplification in less than 38 cycles were considered positive.

### 2.3. Molecular Identification of Paracoccidioides sp. in Soil Samples

To evaluate the presence of *Paracoccidioides* sp. in the soil and provide additional evidence for autochthonous infections in the Tocantins-Araguaia basin, we collected soil samples from armadillos’ burrows and performed a nested PCR assay [21]. Fifteen soil samples were collected in two different locations within Araguaína-TO municipality on 17–18 of November 2019. After identifying an armadillo burrow, we used a shovel adapted with a metal rod to scrape the inner part of the burrow. About 50 g of soil was collected and stocked in 50 mL sterile containers. The shovel was decontaminated with ethanol 70% between each sampled burrow. Environmental DNA was extracted using the DNeasy PowerSoil Kit (QIAGEN, Germany) according to the manufacturer’s instructions. About 0.5 g of soil was used as input material and vigorously agitated on a Precellys homogenizer (Bertin, France). Ten cycles of one minute at 6500 rpm were used for each batch of samples. DNA was quantified in a NanoVue spectrophotometer (GE Healthcare, Chicago, IL, USA) and was normalized to a final concentration of 10 ng/μL. PCR reactions were carried out using the GoTaq Master Mix (Promega) in a final volume of 25 μL, primers at a final concentration of 0.5 µM, and 30 ng of DNA template in a Veriti 96-well (Applied Biosystems, Waltham, MA, USA) thermal cycler. The first amplification reaction was performed using the ITS4 and ITS5 as outer primers [22] using the following steps: 95 °C for 3 min; 35 cycles of 95 °C for 40 s, 55 °C for 50 s, 72 °C for 50 s; 72 °C for 5 min. Three microliters of the first PCR reaction were used as the template for the second *Paracoccidioides*-specific PCR. The final amplification reaction was performed using the inner primers ParIntS (5′-CACGTTGAACTTCTGGTTCG-3′) and ParIntR (5′-TGTCGATCGAGAGAGGAACC-3′), using the following steps: 95 °C for 1 min; 30 cycles of 95 °C for 20 s, 57 °C for 30 s, 72 °C for 40 s; 72 °C for 5 min. The armadillos’ burrows were considered positive by verifying a 403 bp PCR product on a 1.5% agarose gel electrophoresis stained with 0.5 μg/mL of ethidium bromide. To confirm the specificity of the amplicons, one PCR product was sequenced using the Sanger method in an ABI 3500 instrument (Applied Biosystems) using the BigDye Terminator v3.1 sequencing kit. The sequenced amplicon was blasted against the nucleotide collection (nr/nt) using the blastn suite [23].

### 2.4. Georeferencing of PCM Cases

Environmental and clinical data, along with samples genotyped via qPCR, were further georeferenced with QGIS 3.10.9 software (QGIS Development Team 2020, http://www.qgis.org/, accessed on 10 October 2021) using Geographic Coordinate System, SIRGAS 2000 Datum. The cartographic base used, referring to territorial structures, was obtained from the Brazilian Institute of Geography and Statistics (IBGE 2020).

## 3. Results

### 3.1. Clinical Characteristics of PCM in Araguaína

A total of 327 biological samples from 240 patients were collected, including 190 sputum, 113 bronchoalveolar fluid, 9 shaved skin lesions, 6 pleural fluid, 4 lymph node biopsies, 2 palate fragments, 1 bone secretion, 1 abscess secretion, and 1 orotracheal secretion. We confirmed 106 (44.2%) paracoccidioidomycosis cases by direct microscopic examination with 30% KOH including 59 sputum (55.6%), 41 BAL (39%), 1 shaved skin lesion (0.9%), 1 pleural fluid (0.9%), 1 lymph node biopsy (0.9%), 1 palate fragment (0.9%), 1 bone secretion (0.9%), and 1 abscess secretion (0.9%). The morphology of the fungal structures deduced by optical microscopy using KOH staining was compatible with *Paracoccidioides* sp. presenting single to multiple budding round yeast cells with birefringent cell walls (Figure 1A,B).

Demographic and occupational data related to gender are shown in Table 1. We recovered only one *Paracoccidioides* sp. isolate from all tested samples.

The *Paracoccidioides* sp. isolate (PbTO) also displays single to multiple budding yeast cells with birefringent cell wall as deduced by confocal microscopy (Figure 1C,D). Of the total number of confirmed cases, 77 (72.6%) were male and 29 (27.4%) were female (male:female ratio–2.65:1). The age group most affected by PCM was greater than 50 years of age (59.4% of cases), and the overall mean age was 53.92 ± 15.76 years, with a minimum of 17 and a maximum of 87 years. The proportion of females and males was 48.3% and 37.7% for <50 years as well as 51.7% and 62.3% for ≥50 years, with a mean of 50.27 ± 14.75 (minimum 29 and maximum 87 years) and 55.29 ± 16.01 years (minimum of 17 and maximum of 86 years), respectively. The chronic clinical form prevailed in 71 cases (67%) and was similarly prevalent in both genders. The acute form of the disease was observed in 35 cases (33%). The main occupation of patients with PCM was farmer (67%) followed by housewife (16%), salesperson (4.7%), machine operator (2.8%), professional driver (1.9%), and student (1.9%), and the remaining activities by 0.95% of patients each. Among women, the occupational activity of housewife was the most prevalent, with 58.6% of cases, followed by agricultural occupation or farmers (34.5%). Most of the cases were diagnosed in patients living in Araguaína, Tocantins state, and other cities in the northern part of the state (78.3%—Figure 2).

The remaining cases were diagnosed in patients living in Pará (17.92%), Maranhão (2.83%), and Mato Grosso (0.95%) states (Figure 2). PCM/tuberculosis was reported for four cases (3.8%), PCM/HIV and PCM/cancer for three cases each (2.8%), PCM/cutaneous leishmaniasis for two cases (1.9%), and PCM/malaria and PCM/leprosy for one case each (1.4%) and were usually unrelated to gender even though some cases could be biased, such as cervical cancer. One patient (0.9%) died by the time the study was conducted.

### 3.2. Molecular Identification of Paracoccidioides sp.

Using real-time PCR, we detected the presence of *Paracoccidioides* in 37 samples (Table 2). The *gp43* gene was detected in 21 (57%) samples, while the *hsp70* gene was positive in 11 (30%) samples, and both species were detected in 5 (13%) samples. Fourteen samples were negative for the qPCR assay. In addition, the PbTO strain was also subjected to qPCR screening, indicating that this strain belongs to the genus *Paracoccidioides*.

### 3.3. Environmental Detection of Paracoccidioides sp. in Araguaína

Since most PCM cases were detected in patients living in Araguaína and nearby areas, we collected soil samples from armadillos’ burrows to confirm by a Paracoccidioides-specific nested PCR that local infections are likely to occur. We collected 15 samples in two different areas in Araguaína, herein referred to ARA1 (seven samples) and ARA2 (eight samples) (Figure 3). We identified four samples positive for ARA1 (57.14%) and five samples positive for ARA2 (62.5%), suggesting that *Paracoccidioides* sp. is found in the environment, reinforcing that PCM autochthonous cases occur in Araguaína (Figure 3).

We sequenced and deposited one positive amplicon from ARA2-1 site (NCBI accession number MZ424096) and verified that this amplicon has 100% of identity to *P. brasiliensis*, including soil and aerosol samples collected in surrounding areas of Goiás (KP636448) and Mato Grosso states (MF078064).

## 4. Discussion

Campbell [24] predicted that PCM would become a growing threat as humans increasingly disturb the soil in areas where the fungus is native, as occurred in Tocantins, Maranhão, and Pará States, which had been colonized during the last five decades. Studies have shown that expansion of the population associated with the construction of highways and training of workers, especially for rural activities, affects native forests and leaves the soil unprotected, increasing PCM cases [25,26,27]. It is worth mentioning that the MATOPIBA agricultural frontier is on the rise, increasing the risk of PCM infections due to intense deforestation and agricultural activities [28,29].

In our study, we identified 106 new cases of PCM disease in 27 months (47.1 cases/year) through direct mycological examination. Considering that the direct mycological examination has low sensitivity [19,30], these cases may represent only a part of severe PCM cases. The state of Rondônia, located in the Western Brazilian Amazon, is considered a hyperendemic area of PCM, whereas 52.7 cases/year were described [31]. In a case study in Mato Grosso do Sul state, an average of 22.2 cases/year was reported, similar to Maranhão, showing 21.6 cases/year [25]. In Southern Brazil, an average of 25.6 cases/year in a case series of 1000 patients was reported [30]. These studies demonstrate that Tocantins is a PCM hotspot with higher endemicity than other endemic or hyperendemic regions.

According to Zancopé et al. [19], the identification of fungal structures in samples with suspected mycosis is a technique that requires considerable experience and may result in false-negative diagnosis due to the low sensitivity. Mycological culturing was also challenging due to small quantities of fungal structures, patients under empirical antifungal therapy, and longtime culture contamination risks [6], which explains our single isolate. However, for the first time, we included *P. lutzii* as an endemic pathogen to the Tocantins states as deduced by qPCR analysis (Table 2).

In our demographic findings, PCM was prevalent in males greater than 50 years old, similar to other published studies [25,30,32]. However, the ratio between males and females (2.65:1) varied by region and was similar to what was described in Paraná State [33], but differs from those in Rio Grande do Sul (27.3:1) [34], Mato Grosso do Sul (10:1) [35], and Maranhão (8.4:1) [25]. According to Severo et al. [34], the low gender ratio observed in our study suggests that women are beginning to adopt behaviors exposing them to the fungus, such as smoking, alcoholism, and other drugs, and performing labor activities previously performed only by men. Another noticeable characteristic was the high number of cases in women over 50 years old; post-menopausal women have the same susceptibility as men [34,36]. This observation can be explained by estrogen causing blockage or delay in the transition from mycelium to yeast that occurs after infection by the fungus [37,38]. However, we cannot rule out the issue of duration of exposure to the agent among older patients, which could result in a cumulative effect and be a confounding factor in interpreting these results.

The chronic clinical form prevailed between the two genders in most individuals 50 years or older, and as expected, the acute form predominated among individuals less than 50 years old. Our findings are consistent with data from other studies, where 74.6% of cases in São Paulo [30] and 77.4% in Brasilia [32] were characterized as a chronic form. A higher rate of chronic cases, 91.1%, was found in Paraná [33].

Regarding the origin of the patients, the chronic clinical form was predominant in the states of Tocantins (71%), Pará (52%), and Mato Grosso (100%), and the acute clinical form only predominated in patients originated from the state of Maranhão (67%). Reported data on this state also presented higher numbers of clinical cases in the frontier to Tocantins [25]. This result may also suggest that transmission is occurring more intensely in that region, but data for comparative epidemiology and their climate evolution are lacking, complicating drawing any conclusion.

Most cases were related to occupation and linked with rural origin and the chronic clinical form. Several studies indicate that PCM is more prevalent among rural workers and farmers, which was also observed in males in our research [2,12,37]. This result is likely due to reduced access to health care, leading to delays in diagnosis and treatment and the consequent worsening PCM cases. Another issue is the poor health surveillance among rural workers, and since PCM is considered an occupational disease, further monitoring and research are sorely needed [2,39]. However, we observed a lower frequency of rural occupation among women, demonstrating possible cases of urbanization [13]. These findings differ from those postulated in previous literature discussions, indicating the urgent need for further epidemiological studies [34].

Tocantins is located in a region considered by the literature to be of low to moderate endemicity for PCM [2,8]. Although most cases (nearly 80%) are residents of Tocantins (Figure 2), we cannot confirm that all cases are autochthonous cases because there is a substantial constant migration between Northern and Midwestern Brazil. However, the endemicity of PCM in this region was confirmed by molecular analysis of soil samples collected from armadillo`s burrows (Figure 3), suggesting that local infections are likely to occur. Araguaína and the surrounding areas present ideal climatic and geological conditions for the growth of *Paracoccidioides* spp. [40,41].

Diagnosis via molecular methods has been incorporated into clinical laboratories’ routines to enhance efficiency in diagnosing these infections. Our present work is promising since using a fast and accurate qPCR-based test, we can detect the etiological agents of PCM using biological samples. Unfortunately, the primers used for the qPCR assay in this work cannot discriminate different *Paracoccidioides* species, and further analyses are sorely needed to precisely identify the genetic background of the *Paracoccidioides* genotype in this hyperendemic area. However, according to our amplicon sequencing, we observed the presence of at least *P. brasiliensis* genotype in Araguaína, suggesting that this species is endemic to the Tocantins state. However, since *P. lutzii* was found in neighbor states of Tocantins (i.e., Goiás and Mato Grosso), we cannot rule out that this species can also be found in this state. Different genotypes may impact the disease outcome of these patients since different species might display different virulence or pathogenesis traits.

## 5. Conclusions

In this work, we ranked the Tocantins-Araguaia basin as a novel hyperendemic area of PCM in Brazil. Due to the lack of information about PCM, public health administrators have been advocating that this disease would be included in the national compulsory disease notification system, which would be a major breakthrough in the epidemiology, control, and treatment of this important endemic mycosis. In this study, we demonstrated that the migration of new human populations into uninhabited regions must meet new criteria for public and occupational health, posing a challenge to the public health system, which will also require new interfaces and extensive collaboration with previously established centers.

## Figures and Tables

**Figure 1 jof-08-00502-f001:**
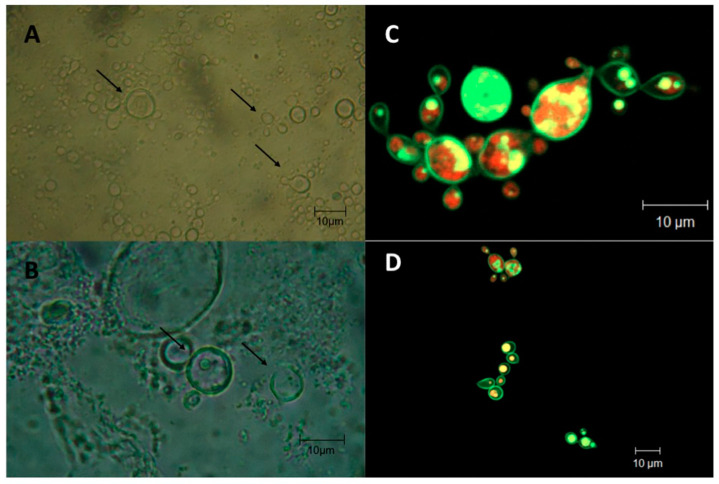
Morphological characteristics of *Paracoccidioides* sp. yeast cells recovered from direct optical microscopy exam (**A**,**B**) and confocal microscopy of the PlTO strain (**C**,**D**). Arrows indicate the presence of single or multibudding yeast cells.

**Figure 2 jof-08-00502-f002:**
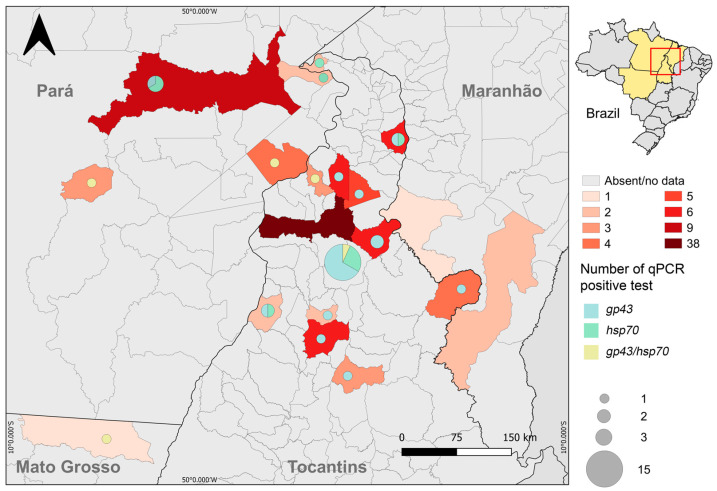
Spatial distribution of paracoccidioidomycosis cases in the Tocantins-Araguaia basin (red frame). Darker colors indicate a wider number of PCM cases. The PCM cases detected by qPCR were plotted next to each respective municipality in a pie chart. *Paracoccidioides* sp. infections detected by qPCR based on *gp43*, *hsp70*, or both markers are represented by different colors.

**Figure 3 jof-08-00502-f003:**
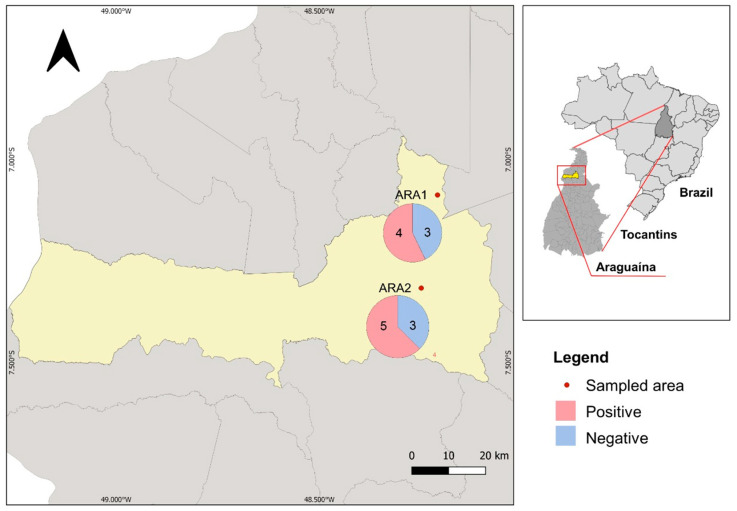
Environmental detection of *Paracoccidioides* sp. in Araguaína-TO. The location of each sampled area is pointed out on the map, and the number of positive and negative samples for the nested PCR assay are represented in a pie chart next to each sampled location.

**Table 1 jof-08-00502-t001:** Demographics, occupation, and comorbidities of patients with paracoccidioidomycosis in Araguaína Tocantins sorted by gender.

Gender		Total *n* = 106	Female *n* = 29	Male *n* = 77	*p*
Clinical form	Acute	35 (33%)	13 (44.8%)	22 (28.6%)	NS
	Chronic	71 (67%)	16	55	
Age	<50 years	43	14 (48.3%)	29 (37.7%)	NS
	≥50 years	63	15 (51.7%)	48 (62.3%)	
Origin	Maranhão	3	2	1	NS
	Mato Grosso	1	0	1	
	Pará	19	5	14	
	Tocantins	83 (78.3%)	22 (75.9%)	61 (79.2%)	
Occupation	Political advisor	1	0	1	*p* < 0.05
	Housewife	17 (16%)	17	0	
	Student	2	0	2	
	Manager	1	0	1	
	Farmer	71 (67%)	10	61	
	Driver	2	0	2	
	Machine operator	3	0	3	
	Locksmith	1	0	1	
	General services	1	1	0	
	Electronic technician	1	0	1	
	Technical nursing	1	1	0	
	Salesperson	5	0	5	
Exposure (*)	In-house activities	19	17	2	*p* < 0.001
	Outdoor activities	87	12	75	
Comorbidities	Cancer	3	3	0	NS
	Leprosy	1	0	1	
	HIV	2	0	2	
	HIV/Tuberculosis	1	0	1	
	Cutaneous Leishmaniasis	2	0	2	
	Malaria	1	1	0	
	Tuberculosis	3	0	3	
	No coinfection	93	25 (86.2%)	68 (88.3%)	

HIV: human immunodeficiency virus. NS: not significant. (*) Exposure was deemed the environment of the main activity, e.g., housewife and student primarily indoors and the remaining activities primarily outdoors.

**Table 2 jof-08-00502-t002:** Demographics of patients with paracoccidioidomycosis in Araguaína, Tocantins, sorted by gene detected.

		Total Positive for qPCR Assay *n* = 37	*gp43* Marker *n* = 21	*hsp70* Marker *n* = 11	*gp43*/*hsp70* Marker *n* = 5	
Age	<50 years	14	6	5	3	NS
	≥50 years	23	15	6	2	
Gender	Female	8	4	3	1	NS
	Male	29	17	8	4	
Origin	Mato Grosso	1	0	0	1	*p* < 0.05
	Pará	7	1	4	2	
	Tocantins	29	20	7	2	

## Data Availability

The data presented in this study are openly available in NCBI, accession number MZ424096.

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
