# Peer review of "Clinical and Eco-Epidemiological Aspects of a Novel Hyperendemic Area of Paracoccidioidomycosis in the Tocantins-Araguaia Basin (Northern Brazil), Caused by Paracoccidioides sp."

_jof, 2022, doi:10.3390/jof8050502_

Round 1

Reviewer 1 Report

The manuscript described the clinical and eco-epidemiological aspects of a novel hyperendemic area of paracoccidioido mycosis in the Tocantins-Araguaia basin, caused by Paracoccidioides sp. The research has scientific rigor and the methods used to respond to the objectives have been carried out properly.

Why authors did not consider amplifying and sequencing the ITS region to identify the species of the fungus?

For improving the manuscript, here are the following corrections or suggestions:

Lines 4, 28, 228, 234, 312. Why do authors use sp instead of spp? Please homogenize it throughout the writing.

Line 42. Include a comma after P. venezuelensis

Lines 206, 215. “gp43” and “hsp70” in italics

Lines 132, 139. 50mL, 25mL

Line 138, 139. 10ng/uL, 25uL

Lines 233, 234. (57.14%), (62.5%)

Line 273. Severo et al.

Please, include a Conclusion section.

Author Response

The manuscript described the clinical and eco-epidemiological aspects of a novel hyperendemic area of paracoccidioidomycosis in the Tocantins-Araguaia basin, caused by Paracoccidioides sp. The research has scientific rigor and the methods used to respond to the objectives have been carried out properly.

A: We appreciate the comments and specific comments are listed point-by-point below

Why authors did not consider amplifying and sequencing the ITS region to identify the species of the fungus?

A: ITS amplification and sequencing only allows the discrimination between P. brasiliensis complex and P. lutzii. Unfortunately this isolate was lost but new efforts in order to precisely identify novel stains using whole genome sequencing will be further carried out.

For improving the manuscript, here are the following corrections or suggestions:

Lines 4, 28, 228, 234, 312. Why do authors use sp instead of spp? Please homogenize it throughout the writing.

A: According to the species nomenclature rules, "sp." is an abbreviation for species and is used when the actual species name cannot or is not necessarily be specified. The plural form of this abbreviation is "spp." and is used to indicate "several species”. In this context, we decided used sp. or spp. when needed. Unfortunately, we were only able to resolve the genotype using the sequencing of the soil ITS amplicon but not the isolate. We fixed the right species abbreviation in line 234.

Line 42. Include a comma after P. venezuelensis

A: This was included accordingly

Lines 206, 215. “gp43” and “hsp70” in italics

A: This was modified throughout the manuscript.

Lines 132, 139. 50mL, 25mL

A: This was modified throughout the manuscript.

Line 138, 139. 10ng/uL, 25uL

A: This was modified throughout the manuscript.

Lines 233, 234. (57.14%), (62.5%)

A: This was modified accordingly.

Line 273. Severo et al.

A: This was modified accordingly.

Please, include a Conclusion section.

A: We have included a conclusion section. Please see lines 329-337

Reviewer 2 Report

There is no major suggestion in the manuscript.

Please consider the suggested parts as follows:

Introduction

Paracoccidioides brasiliensis, P. americana, P. 41 restrepiensis, P. venezuelensis, and P. lutzii, are the etiologic agents of PCM [3,4], although P. loboii and P. cetii were added recently for the causative agents of paracoccidioidomycosis loboi and paracoccidioidomycosis ceti, respectively. [Please, add the reference; Vilela R, Huebner M, Vilela C, Vilela G, Pettersen B, Oliveira C, Mendoza L. The taxonomy of two uncultivated fungal mammalian pathogens is revealed through phylogeny and population genetic analyses. Sci Rep. 2021;11:18119.]

Results

L215: gp43 should be in Italic.

Author Response

There is no major suggestion in the manuscript.

Please consider the suggested parts as follows:

A: We appreciate the comments. See below:

Introduction

Paracoccidioides brasiliensis, P. americana, P. 41 restrepiensis, P. venezuelensis, and P. lutzii, are the etiologic agents of PCM [3,4], although P. loboii and P. cetii were added recently for the causative agents of paracoccidioidomycosis loboi and paracoccidioidomycosis ceti, respectively. [Please, add the reference; Vilela R, Huebner M, Vilela C, Vilela G, Pettersen B, Oliveira C, Mendoza L. The taxonomy of two uncultivated fungal mammalian pathogens is revealed through phylogeny and population genetic analyses. Sci Rep. 2021;11:18119.]

A: We have included the sentence above as suggested. We appreciate the comments

Results

L215: gp43 should be in Italic.

A: This was modified accordingly